# Present and Future Contributions of Reactor Experiments to Mass Ordering and Neutrino Oscillation Studies

**Vito Antonelli \*, Lino Miramonti**  **and Gioacchino Ranucci**

Dipartimento di Fisica, Università degli Studi di Milano e INFN, Sezione di Milano, via Celoria 16,
I-20133 Milano, Italy; lino.miramonti@mi.infn.it (L.M.); gioacchino.ranucci@mi.infn.it (G.R.)
\* Correspondence: vito.antonelli@mi.infn.it

**Abstract:** After a long a glorious history, marked by the first direct proofs of neutrino existence and of the mixing between the first and third neutrino generations, the reactor antineutrino experiments are still well alive and will continue to give important contributions to the development of elementary particle physics and astrophysics. In parallel to the SBL (short baseline) experiments, that will be dedicated mainly to the search for sterile neutrinos, a new kind of experiments will start playing an important role: reactor experiments with a "medium" value, around 50 km, of the baseline, somehow in the middle between the SBL and the LBL (long baselines), like KamLAND, which in the recent past gave essential contributions to the developments of neutrino physics. These new medium baseline reactor experiments can be very important, mainly for the study of neutrino mass ordering. The first example of this kind, the liquid scintillator JUNO experiment, characterized by a very high mass and an unprecedented energy resolution, will soon start data collecting in China. Its main aspects are discussed here, together with its potentialities for what concerns the mass ordering investigation and also the other issues that can be studied with this detector, spanning from the accurate oscillation parameter determination to the study of solar neutrinos, geoneutrinos, atmospheric neutrinos and neutrinos emitted by supernovas and to the search for signals of potential Lorentz invariance violation.

**Keywords:** reactor neutrinos; neutrino mass ordering; neutrino oscillation; neutrino mass models

## 1. Introduction: Milestones of Reactor Antineutrino Experiments

The nuclear power plants are an ideal source of pure and intense electron antineutrino ($\bar{\nu}_e$) beams, emitted in the radioactive decays of the fissile products of the nuclear fission. The flux of these antineutrinos is, in general, relatively well known, even if their evolution in time is not so easy to keep under control.

The antineutrinos interaction, with the protons of the detector material they cross, takes place mainly through the "inverse $\beta$ decay", that represents the golden channel for the experimental analysis: $\bar{\nu}_e + p^+ \rightarrow e^+ + n$. The study of this process offers a series of advantages: the cross section is not too small (making possible the collection of a statistically significant number of events in case of detectors with high masses) and, above all, the process is characterized by a well distinguishable experimental signature. The final state positron, crossing the material, induces atomic electrons excitation and in some materials, like the ones used in scintillation detectors, subsequent disexcitation with light emission that can be collected by a series of photomultipliers surrounding the detector. The separation between the signal and the main background, represented by the charged particles emitted in the natural radioactive decays taking place in the detector itself, is made easier by the simultaneous

emission (in the inverse $\beta$ decay) of a neutron which, after a few microsecond of migrations inside the detector, can be captured by protons, generating a deuterium nucleus with the emission of a photon with a characteristic energy, 2.2 MeV. Therefore, it is possible to perform an efficient uncorrelated background rejection through positron and neutron signal time coincidence. By the study of the signals and in particular of the positron visible energy, one can also extract information about $\bar{\nu}_e$ energy.

For these reasons, the reactor antineutrino experiments played since ever an important role in the development of our knowledge of neutrino properties. The first great success of this kind of experiments was obtained in 1956, when Cowan and Reines [1,2], with their experiment at the Savannah River nuclear power plant, confirmed the hints by a previous (1953) experiment at the Hanford nuclear power plant and found the first experimental evidence of neutrino (antineutrino in their case) existence, almost thirty years after the visionary theoretical hypothesis by Pauli (and the Fermi's quantitative theory of beta decay).

The first reactor experiments used to study neutrino properties were the so-called short baseline experiments (SBL). For the first of them, operating during the 1980s and the 1990s [3–8], the baseline (distance between the reactor and the detector) was of the order of a few tenths of meters. The baseline was extended to distances of the order of about 700 m for Palo Verde experiment [9] and about 1 km in the case of CHOOZ [10].

All of these experiments did not find any clear evidence of neutrino oscillation (due to their limited sensitivity) but they were very important historically to put stringent constraints on mass and mixing parameters; the upper limit on the mixing angle between the first and the third neutrino generation ($\theta_{13}$) was fixed for many years by the short baseline reactor experiments (mainly [9,10]). As a matter of fact, hints of a possible events deficit, with respect to the theoretical predictions in absence of oscillation, seem to come by more recent reanalyses of the first SBL reactor experiments data, in the light of new more precise determination of antineutrino fluxes [11,12], that predicted a systematic increase of the flux above 2 MeV. This "reactor antineutrino anomaly" [13] stimulated a wide debate in literature and might be interpreted also as a possible hint in favor of the existence of an additional light sterile neutrino[1], even if it is very difficult to build an oscillation solution that accomodates both the reactor antineutrino anomaly and other possible indications in favor of the sterile neutrino hypothesis coming from the accelerator sector. It's also important to stress the fact that, in any case, significant residual uncertainties still affect the theoretical calculation of the detailed contributions to the reactor antineutrino fluxes [16–18].

An important milestone of reactor experiments history, which had a great impact on all neutrino physics, has been the study of the 1–2 mixing sector performed, since the first years of the new millennium, by KamLAND [19]. This experiment, that used a 1kton liquid scintillator detector located at the enlarged Kamiokande site, was a long-baseline (LBL) experiment, with a medium distance (from the 51 different reactors producing the $\bar{\nu}_e$ flux to the detector) of the order of 200 km. About 78% of this flux came from 6 reactors, covering a baseline range of 139–214 km. Thanks to these high values of the baseline L, KamLAND could access (even with values of the energies of a few MeV typical of reactor antineutrinos) to the region of the oscillation parameters ($\Delta m^2$ of the order of $10^{-4}$–$10^{-5}$ eV$^2$) that is relevant for the solar neutrinos and was not accessible to the previous SBL reactor experiments. Hence, the KamLAND data, starting from the first 2002 results [20] that showed a deficit of almost 40% of the experimental number of events with respect to the expected theoretical predictions, gave a fundamental independent confirmation of the oscillation hypothesis, with a flux relatively well known and under control, that was not affected by the possible uncertainties present in the solar neutrino flux. Moreover, the combined statistical analysis of KamLAND results and of the ones obtained by the different solar neutrino experiments (and mainly by SNO [21–24]) made possible a better

---

[1]　For a discussion about the possible connection between the "reactor neutrino anomaly" and neutrino oscillation in presence of an hypothetical sterile neutrino, see, for instance [14,15].

determination of $\theta_{12}$ and mainly $\Delta m_{21}^2$ [25], definitely proving the validity of the solution to the Solar Neutrino Puzzle based on the Large Mixing Angle, with the MSW interaction with matter [26–29], as we are going to discuss later in the Section 4.4. For a detailed discussion of KamLAND data impact on the final solution of the "Solar Neutrino Puzzle", we refer the interested reader to the following papers, published immediately after the first KamLAND data [30–38].

About ten years later, between 2011 and 2012, another result of great impact, not only for neutrino physics, was recovered by reactor neutrino experiments, namely by three short baseline (SBL) experiments: Daya Bay [39], Double CHOOZ [40] and RENO [41] performed the first direct experimental measurement of $\theta_{13} \neq 0$. In the following years these experiments improved their accuracy, offering a more and more precise determination of $\theta_{13}$ and contributing also to the knowledge of $\Delta m_{32(1)}^2$. The main achievements and the perspectives of these three SBL reactor experiments and the impact of their results will be discussed in the next section.

After this, we will also discuss, in Section 3, another issue central in the present and future of reactor neutrino physics, that is the opportunity to attack the mass ordering puzzle and to investigate a series of other interesting central topics of elementary particle physics and astrophysics by means of medium baseline experiments (with values of L around 50 km). We will focus in particular our attention on the JUNO experiment [42], that will soon start data taking in China, and in Section 4 we will discuss in detail its main characteristics and potentialities.

Finally we will finish our paper with a critical summary of the main achievements of reactor neutrino physics and a brief discussion of the challenges they could face in the near future.

## 2. Short-Baseline Reactor Experiments

### 2.1. Daya Bay, RENO and Double CHOOZ

The three largest and probably most famous SBL reactor antineutrino experiments [43–45] have some common aspects, that can be summarized in the following way.

- All of the three make use of near and far detectors, that is detectors (the near ones) placed at relatively short distances from the nuclear reactor emitting the antineutrino flux and other bigger detectors (the far ones) settled at larger distances from the source, but with a baseline L that, anyhow, is of the order of 1 km (or similar values). The presence of a near detector is fundamental for a better direct check and monitoring of the reactor flux and because, by comparing the number of events collected at the near and at the far detectors, and considering the natural flux reduction (proportional to $1/R^2$) due to geometrical reasons, one can study the L dependence of the oscillation phenomenon and reduce the systematic uncertainty associated with the partial knowledge of the flux and of its time evolution. The addition of the near detector has guaranteed a significant increase in the statistical significance of the results obtained by all the three SBL experiments.
- All of these experiments are designed as a nested structure with three main parts:

  (a) An internal Gd-LS (Gadolinium-loaded liquid scintillator) detector, that is a liquid scintillator, acting as $\bar{\nu}_e$ target, with the addition of a relatively small quantity (0.1% by mass in the Daya-Bay case) of Gadolinium, having the purpose to increase the neutron capture rate, essential for the inverse $\beta$ decay study. The presence of Gadolinium determines a significant shortening of the neutron-capture time, which is reduced to values around 30 μs as compared to ~200 μs, typical for a liquid scintillator. This reduces the accidental background rate by almost one order of magnitude.

  (b) A pure liquid scintillator, which is useful to increase the resolution and guarantees a better energy measurement

(c)　An external mineral oil radioactivity shield, with the function to reduce as much as possible the impact of the natural radioactivity background and improve the signal to background ratio.

- In all the cases the detector is surrounded by an array of photomultipliers and an external water pool, acting as a shield and cosmic ray detector.

Despite their similarities, the three SBL detectors differ from each other for important specific aspects, among which the total reactor thermal power, the target mass and the overburden, that are summarized in Table 1. One can observe, for instance, that the Daya Bay detector has a much larger (almost a factor ten) value of the mass with respect to the other two experiments and mainly to Double-CHOOZ. This reflects, obviously, in a significantly higher statistics for the Chinese experiment, that takes advantage also from an higher value of the overburden (more than eight hundred meters of water equivalent for the far detector).

**Table 1.** Comparison of the main parameters of Daya Bay, Double CHOOZ and RENO. Table taken from [46].

|  | $P_{th}$ (GW) | Target Mass at Far Site (tons) | Overburden (Near/Far) (mwe) | Data Taking (Start-End) |
|---|---|---|---|---|
| Double CHOOZ | 8.6 | 8.3 | 80/300 | 2011–2017 |
| RENO | 16.4 | 15.4 | 90/440 | 2011–2021 |
| Daya Bay | 17.4 | 80 | 250/860 | 2011–2020 |

## 2.2. Recent Results of the SBL Experiments

The main recent contribution of SBL experiments to the knowledge of neutrino physics has been for sure the proof that the mixing angle between the first and third neutrino generations ($\theta_{13}$) is different from zero [39–41]. This result confirmed the previous hints, coming by LBL accelerator experiments [47,48] and global phenomenological analyses [49,50], with a much more robust statistical significance, largely exceeding by today the $5\sigma$ level.

By performing a combined analysis of the observed $\bar{\nu}_e$ rate (looking for oscillation signals) and of the spectral shapes (studying the energy dependence of antineutrino disappearance), the three SBL experiments were able to extract, together with the value for the mixing angle $\theta_{13}$, also the absolute value of the difference between the squared neutrino mass eigenvalues $\left( \left| \Delta m^2_{32(31)} \right| \right)$ [51–53]. The results with their relative uncertainties are reported in Table 2 and in Figure 1, where they are compared also with the values for the same oscillation parameters that can be recovered by the analysis of LBL accelerator (mainly T2K [54] for the mixing angle and also MINOS [55] and NO$\nu$A [56] for $\Delta m^2$) and of atmospheric (SuperKamiokande) [57] experiments.

**Table 2.** Summary of the oscillation parameter values extracted by the analysis of the three main present SBL reactor experiments.

| | Experiment | | |
|---|---|---|---|
| **Parameter** | **Daya-Bay [51]** | **RENO [53]** | **DoubleCHOOZ [52]** |
| $\sin^2(2\theta_{13})$ | $0.0856 \pm 0.0029$ | $0.0896 \pm 0.0048(\text{stat}) \pm 0.0047(\text{syst})$ | $0.105 \pm 0.014$ |
| $\left\| \Delta m^2_{32} \right\|$ $(10^{-3}\text{eV}^2)$ | $2.471^{+0.068}_{-0.070}$ (NH) $2.575^{+0.068}_{-0.070}$ (IH) | | |
| $\left\| \Delta m^2_{ee} \right\|$ $(10^{-3}\text{eV}^2)$ | | $2.68 \pm 0.12.(\text{stat}) \pm 0.07(\text{syst})$ | |

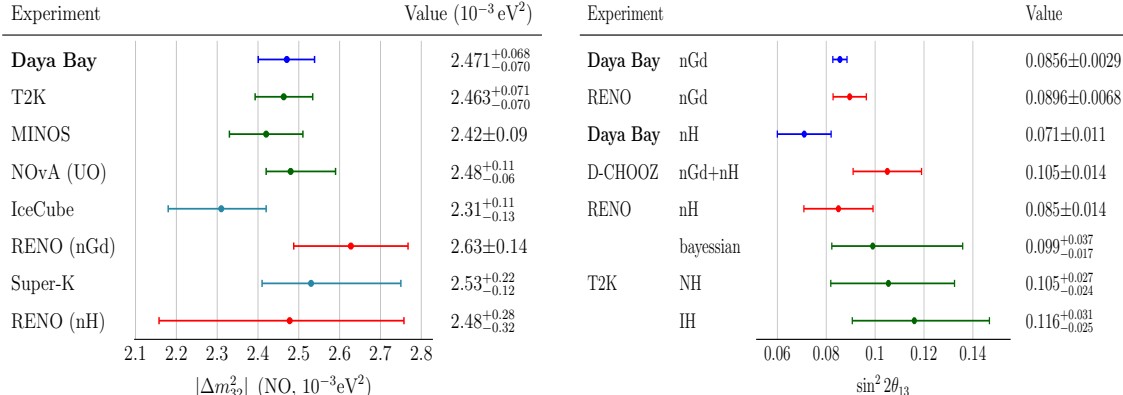

**Figure 1.** Values for the oscillation parameters recovered by the three main present SBL reactor experiments, compared with the results by the LBL accelerator experiments and by SuperKamiokande atmospheric neutrino studies. The $\Delta m_{32}^2$ values refer to the normal ordering case. For Daya-Bay and RENO experiments the data set relatives to neutrons (emitted in the inverse $\beta$ decay) captured by gadolinium (labelled as nGd) [51,53] are reported together with the ones obtained with neutrons captured by hydrogen (labelled as nH) [58,59]. For Double Chooz the data reported refer to the analysis including the total neutron capture detector [52]. Figure obtained by courtesy of M. Gonchar and similar to the one present in [46].

The proof that $\theta_{13}$ is significantly different from zero has been essential, not only to further clarify the mass and mixing pattern, but also to open new possibilities for the experiments searching for leptonic CP violation and the ones investigating the mass ordering, by looking at effects whose amplitude is proportional to $\sin^2(2\theta_{13})$, as we will discuss in the next sections.

*2.3. Open Issues in SBL Reactor Experiments*

One of the main points that still must be clarified in reactor neutrino physics is the presence of the so called "flux anomaly". All the three main SBL experiments have observed a significant deviation from the predicted spectrum (a sort of bump) in the region between 4 and 6 MeV. This anomaly has been recently confirmed also by NEOS (Neutrino Experiment for Oscillation at Short baseline) [60], a short-baseline reactor experiment running since 2015 in Korea, with the main aim to verify the possible existence of a sterile neutrino at the eV mass scale. At present there is no unique final explanation of this "anomaly" and the problem is still under investigation, but the different analyses [61–63] seem to indicate a probable correlation with the reactor power and fuel composition evolution and with the knowledge of the flux from the different fissile products, mainly from the ones generated by $^{235}$U and $^{239}$Pu. Alternative solutions based on a particle origin of the anomaly and predicting non standard interactions are still viable, but less probable [64].

Another interesting issue is the analysis of possible spectrum distortions, in combination with MINOS results. The aim is that of looking for signals of sterile neutrinos and probing the LSND and MiniBoonE anomalies. Up to now no evidences have been found in this direction and this determined a significant reduction of the allowed regions in the parameter space. The values of $\Delta m_{41}^2 < 0.8 \, \text{eV}^2$ are excluded at about 95% C.L.

On the other hand, there have been hints by Daya-Bay, Reno and Double Chooz precise flux measurements of a reactor $\bar{\nu}_e$ events deficit, confirming the reactor antineutrino anomaly already emerging by the old SBL data, as discussed in Section 1. The reason for this discrepancy between the experimental data and the reference theoretical model is still not completely known [65] and this stimulated different attempts of explanation. One possibility is a partial mistake in the currently adopted reactor neutrino flux models, confirmed also by the presence of the "bump" (discussed above) in the region of the spectrum between 4 and 6 MeV and by the partial inconsistency with the fuel

evolution results observed by Daya Bay and RENO. As underlined in [66–68], such a solution for the reactor antineutrino anomaly would require mainly a theoretical revaluation of the $^{235}$U reactor antineutrino flux.

Another possible explanation, that could even be complementary to the first one, would be the existence of an eV-scale light sterile neutrino. This hypothesis is under investigation by many experiments, running at present or planned for the near future (in addition to the cited NEOS experiment [60], we can remember STEREO [69,70] in France, DANNS [71] and Neutrino-4 [72,73] in Russia, PROSPECTS [74] in the United States and many other experiments[2]. The first results seem to restrict the eventual possible values to higher $\Delta m^2$.

### 2.4. Future of Reactor Neutrino Experiments: From SBL to Medium-Baseline Experiments

A change of paradigm is taking place in these years in reactor neutrino physics. As discussed in the previous part of this paper, apart from the long-baseline KAMLAND experiment, essential to confirm the solution of the Solar Neutrino Puzzle and extract the mass and mixing parameters of the 1-2 sector, the main achievements of this research field were obtained by short-baseline experiments, with the proof of neutrino existence and of the fact that the first and third neutrino generations are not decoupled, the determination of the values of $\theta_{13}$ and the contribution to the determination of $\left|\Delta m^2_{32(31)}\right|$. This kind of experiments will continue also in the next years to play a relevant role in neutrino physics, as discussed in Section 2.3, but, meanwhile, a new sector of analyses is going to be explored in the very near future and this will require the use of different kind of reactor antineutrino experiments, characterized by medium values of the baseline (around 50 km), huge detector masses and very high energy resolution. This kind of experimental apparatus will make possible the study of one of the main present open questions of neutrino physics, that is the determination of neutrino mass ordering. At the same time they are naturally multipurpose experiments, enabling to investigate many other topics relevant both for elementary particle physics and for astrophysics.

All of these items will be discussed in the rest of the paper. We will start in Section 3 from the concept of neutrino mass hierarchy, the importance of its determination in the general theoretical framework of neutrino mass and mixing and the possibility for reactor antineutrino experiments to contribute significantly to this kind of studies. Then, in Section 4, we will focus on the experiment JUNO, that offers the first important example of detector designed to perform the kind of research project we just discussed and that will start its data taking very soon in China.

## 3. Reactor Neutrino Experiments and Neutrino Mass Ordering

### 3.1. The Neutrino Mass Ordering

Sixty years after the revolutionary Pontecorvo's idea [77,78] that neutrinos are massive and oscillating particles, many steps forward have been done in our knowledge and by now, thanks to the data obtained by a pletora of disappearance and appearance experiments (using different neutrino and antineutrino beams, produced by various natural and artificial sources, covering a wide range of energies), not only we have the proof that Pontecorvo was right, but we have also been able to reconstruct in a quite clear way the general pattern of mass and mixing framework. It is clear that, even leaving apart for the time being the hypothesis of additional sterile neutrinos, there are three different neutrino mass eigenstates, with quite well known values of the two differences between the squared mass eigenvalues ($\Delta m^2_{21}$ and $|\Delta m^2_{31(32)}|$), separated by almost two orders of magnitude; moreover the three mixing angles are known with a quite satisfactory accuracy, of the order of a few

---

[2]  For a discussion about future reactor experiments looking for sterile neutrino see also the talk [75] given by T. Lasserre at the *European Neutrino Town Meeting* (Cern, October 2018) and the report [76].

percent, with significant possibilities of improvement for near future experiments (as will be discussed in the Section 4.3).

Nevertheless, many important open questions in this field are still waiting for an answer. Some of them, like the determination of neutrino nature (Dirac or Majorana fermion), could eventually remain unsolved, at least for many years, but there are other important issues that we can hope to understand in a medium (like in the case of the value of the CP violating phase) or even short time scale. This is the case, for instance, of the solution of the so called octant problem (that is the discrimination between values of $\theta_{23}$ lower or higher than $\frac{\pi}{4}$) and, even more relevant, of the solution of the so called hierarchy problem, that is the determination of the exact ordering of the three neutrino mass eigenvalues.

At present, the experimental data are still consistent with two possible ordering for these eigenvalues, both illustrated in Figure 2.

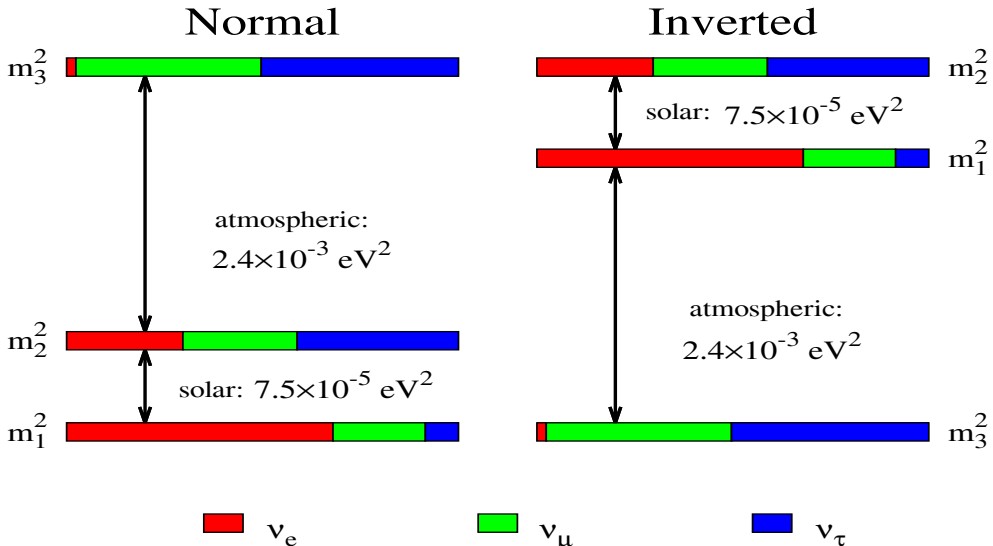

**Figure 2.** The two possible schemes for the neutrino mass eigenvalues: on the left the normal ordering (or normal hierarchy), with $\left|\Delta m^2_{31}\right| = \left|\Delta m^2_{32}\right| + \Delta m^2_{21} > \left|\Delta m^2_{32}\right|$ and, on the right, the inverted ordering (or inverted hierarchy), with $\left|\Delta m^2_{31}\right| = \left|\Delta m^2_{32}\right| - \Delta m^2_{21} < \left|\Delta m^2_{32}\right|$. The two mass eigenvalues squared differences are not represented in scale, but, in any case, one can appreciate the gap existing between the "solar" and the "atmospheric" ones. The different colors represent the flavor compositions of all the three neutrino mass eigenvalues.

The first possibility would correspond to what is usual denoted "normal ordering" , or normal hierarchy (NH), in which the third mass eigenvalue $m_3$ is the highest one, well separated by the other two ($m_3 \gg m_2 > m_1$). However we could also have a so called "inverted ordering", or "inverted hierarchy", corresponding to $m_3$ as the lowest eigenvalue, well separated by the other two higher mass values ($m_3 \ll m_1 < m_2$).

The mass ordering determination is one of the main present issues of neutrino physics for multiple reasons. First of all it will have a direct impact on the potential discoveries of present and future experiments searching for leptonic CP violation and mainly for possible signals of neutrinoless double beta ($0\nu2\beta$) decays. As far as this last issue, one can show that in case of normal ordering the allowed parameter space would significantly be reduced and difficult to access experimentally (mainly for what concerns the value of $m_{\beta\beta}$ which could only be very low). Therefore, we could be in the undesirable condition in which, even if neutrino were a Majorana particle, we could never prove it, at least with present generation of technology and experiments. Moreover, the importance of neutrino mass ordering determination is even more deep, because it could be an essential hint to discriminate between

different possible extensions of the Standard Model and, consequently, it could also help in choosing the better strategy to follow to look for these extensions in the high energy sector.

### 3.2. Present Status of the Mass Ordering Determination

The global analyses [79–81] of neutrino and antineutrino experiments show some indications in favor of the normal mass ordering, coming from different kind of experiments. This issue is under investigation mainly by LBL accelerator experiments, comparing their results also with the ones from SBL reactor data (mainly Day Bay), and by atmospheric neutrino experiments.

Sensitivity to the mass ordering is provided by the analysis of the matter effect in oscillations for neutrinos and antineutrinos, as well as by comparing the oscillations in $\nu_e$ and $\nu_\mu$ channels. For normal (inverted) ordering the neutrino events would be enhanced (suppressed) by the matter effect, whereas for antineutrino events would be exactly the opposite. It is also important to consider the interplay with the CP violation: depending upon the normal (inverted) ordering the matter effect increases (decreases) the $\delta_{CP}$ impact for neutrinos, while the opposite happens for anti-neutrinos. This is more important for NO$\nu$A than for T2K, due to larger matter effects induced by the longer NO$\nu$A baseline.

The LBL data, collected by T2K experiment [82,83] and NO$\nu$A [56,84] and the ones by the atmospheric neutrino experiments, mainly SuperKamiokande (including all the data up to phase-IV) [85,86] and IceCube DeepCore [87], favor the normal ordering with a statistical significal of at least $2\sigma$. However no final conclusions can be drawn at present.

In future this topic will be studied by dedicated experiments with all the above cited kinds of neutrino and antineutrino beams and also exploring new experimental techniques. The accelerator LBL will continue to play a relevant role [88,89], fully exploiting the NO$\nu$A potential and taking advantage by the advent, in less than ten years by now, of DUNE (Deep Underground Neutrino Experiment) [90–92] (with a very long-baseline, $L \simeq 1300$ km, and large matter effects) and T2HK [93] (characterized by smaller matter effects, but large statistics). Very interesting synergies could be exploited also by combing the data obtained by the last two experiments [94]. Another interesting opportunity could be offered in about twenty years from now by the European Spallation Source (ESS) [95], under construction since 2014, which, in parallel to the rich neutron program, should produce a 300 MeV neutrino beam that could be studied at underground far detectors (with two possible baselines $L \simeq 360$ km and $L \simeq 540$ km.)

As far as the atmospheric neutrino studies, they will continue at neutrino telescopes, with the upgrade of IceCube, and eventually with the PINGU project, and with KM3NeT-ORCA [96].

The real novelty in this field is the advent of a new possible way of studying the mass ordering, by means of medium baseline reactor experiments. We will focus our attention on this last category of experiments in the remaining part of the paper.

### 3.3. Reactor Neutrino Physics and Mass Ordering Determination

The significantly different from zero value of the mixing angle between the first and the third neutrino generation, $\sin^2(2\theta_{13}) \simeq 0.08 - 0.09$, implies, as an important by product, the possibility for present and future experiments to look for signals of leptonic CP violation (proportional to $\sin^2\theta_{13}$) and also to investigate the neutrino mass ordering, by studying the corrections to the $\bar{\nu}_e$ oscillation probability sensitive to this ordering. The possibility to perform this kind of studies by the analysis of inverse $\beta$ decays with medium baseline reactor experiments has been proposed for the first time in [97].

As a matter of fact, in the usual 3 flavor analysis, the electron antineutrino survival probability in vacuum is given by:

$$P_{\text{ee}} = 1 - \cos^4(\theta_{13}) \sin^2(2\theta_{12}) \sin^2(\Delta_{21}) - \sin^2(2\theta_{13})[\cos^2(\theta_{12}) \sin^2(\Delta_{31}) + \sin^2(\theta_{12}) \sin^2(\Delta_{32})]. \quad (1)$$

In (1) we denoted by $\Delta_{ij}$ the following combination of the experimental parameters L (baseline) and E (antineutrino energy) and of the neutrino mass eigenvalues $m_i$ and $m_j$:

$$\Delta_{ij} = \frac{\Delta m_{ij}^2 L}{4E} = \frac{(m_i^2 - m_j^2)L}{4E} \ .$$

In order to make the dependence on the neutrino mass ordering more explicit, the oscillation probability of (1) can be written in the following, way [98–100]:

$$P_{ee} = 1 - \cos^4(\theta_{13})\sin^2(2\theta_{12})\sin^2(\Delta_{21}) - \tfrac{1}{2}\sin^2(2\theta_{13})\left[1 - \sqrt{1 - \sin^2(2\theta_{12})\sin^2(\Delta_{21})}\cos\left(2\left|\Delta m_{ee}^2\right| \pm \phi\right)\right] . \quad (2)$$

In (2) $\Delta m_{ee}^2$ represents the quantity $\Delta m_{ee}^2 = \left[\cos^2(\theta_{12})\Delta m_{31}^2 + \sin^2(\theta_{12})\Delta m_{32}^2\right]$ and the "phase factor" $\phi$ is the combination of the 1-2 sector mass and mixing parameters defined by the relations:

$$\begin{aligned}
\sin\phi &= \frac{\cos^2(\theta_{12})\sin[2\sin^2(\theta_{12})\Delta_{21}] - \sin^2(\theta_{12})\sin[2\cos^2(\theta_{12})\Delta_{21}]}{\sqrt{1 - \sin^2(2\theta_{12})\sin^2\Delta_{21}}} \\
\cos\phi &= \frac{\cos^2(\theta_{12})\cos[2\sin^2(\theta_{12})\Delta_{21}] + \sin^2\theta_{12}\cos[2\cos^2(\theta_{12})\Delta_{21}]}{\sqrt{1 - \sin^2(2\theta_{12})\sin^2\Delta_{21}}} \ .
\end{aligned} \quad (3)$$

The sign in front of the $\phi$ term in formula (2) is equal to +1 in case normal mass ordering and −1 for the inverted ordering case. Changing from one to the other neutrino mass ordering corresponds to a change in the sign of this "phase term". The convolution of the oscillation probability with the reactor antineutrino flux and the cross section (properly computed taking into account the experimental efficiency and resolution) gives an expected spectrum as a function of the energy. In Figure 3 this spectrum is represented (with arbitrary units) as a function of $L/E$, the relevant parameter in case of oscillation. The plot is obtained by choosing a baseline L of the order of 50 km, a typical value for the mass ordering studies at reactor neutrino experiments, as we are going to discuss in the following. In presence of oscillation with $\theta_{13} \neq 0$ the graph shows the superposition to the well-known dominating oscillation behavior of fastly oscillating corrections, whose phase depends upon the kind of mass ordering. Hence, a detailed analysis of the experimental spectrum, supported by sufficiently high statistics and energy resolution, can in principle be used to discriminate between the two possible hypotheses (normal and inverted) for the neutrino mass ordering.

In order to perform such an experimental program, it is essential, to select a value of $L/E$, that maximizes the oscillation amplitude and the relative weight of the hierarchy-dependent corrections to the spectrum. In the case of the JUNO experiment [42], that will start data taking in China in the very next years, the medium baseline of 53 km has been chosen in such a way to satisfy this condition, taking into account the reactor antineutrino flux "spectral distribution". This value corresponds to the region of maximum oscillation for the sector involving the first two neutrino mass eigenstates (the so called 1–2 sector), as shown in Figure 4, representing the situation for different present and future reactor experiments. We will discuss the JUNO case in the next section.

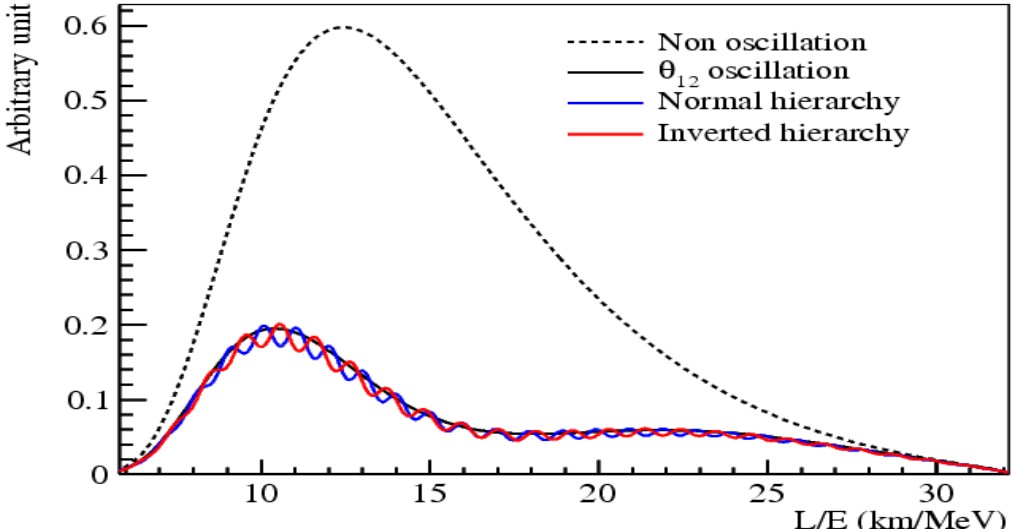

**Figure 3.** Expected reactor antineutrino spectrum, represented as a function of the ratio $\frac{L}{E}$. The curves correspond to the spectrum in absence of oscillation (dotted line), in presence of oscillation but for $\theta_{13} = 0$ (full black), and in the realistic case of oscillation for normal (blue line) or inverted (red curve) neutrino mass ordering. Figure taken by ([101]). Copyright 2008 by the American Physical Society.

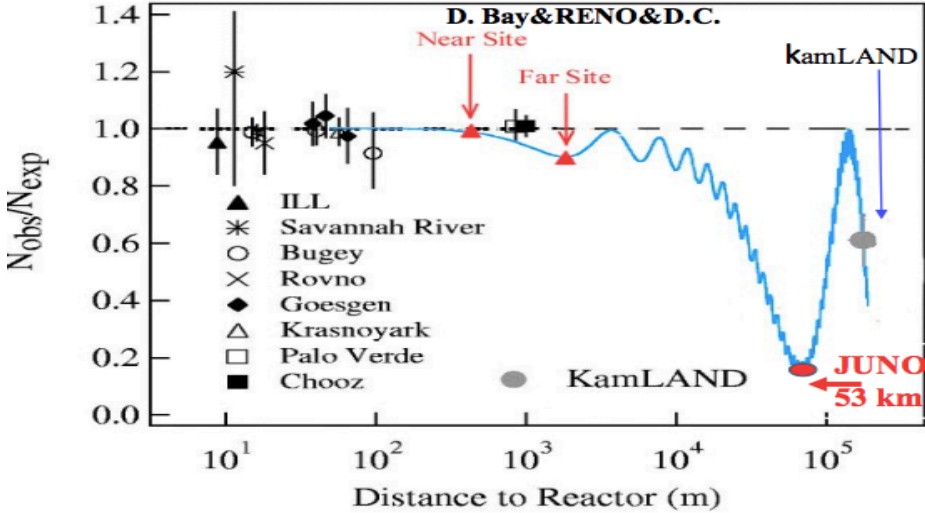

**Figure 4.** Representation of the ratio between the number of observed antineutrino events and the corresponding expected value in absence of oscillation, as a function of the baseline, for different past, present and future reactor experiments. One can notice, among the others, the results of the near and far detectors for Daya-Bay [43], Double CHOOZ [44] and RENO [45] and the ones of KamLAND and JUNO.

## 4. The JUNO Experiment and Its Potentialities

The JUNO (Jiangmen Underground Neutrino Observatory) experiment is a multiple-purpose neutrino experiment, proposed in 2008, approved in 2013 and that will be online in the very next years, near Kaiping in the South of China, at a distance of about 53 km from two different nuclear power plants, at Taishan and at Yangjiang, supplying a total nominal reactor power of about 36 GW.

The main goal of this experiment is the mass ordering study and the precise determination of three of the mass and mixing parameters. In this Section we will discuss in details the JUNO potentialities for what concerns these two aspects. In addition to this, there are many other important studies

that strictly speaking are not directly connected with reactor neutrinos, in the sense that they do not make use of neutrino (or antineutrino) beams produced by nuclear reactors, but can be performed at JUNO, offering a further confirmation of the essential contributions that reactor neutrino experiments have given and continue to give to the development of elementary particle physics and astrophysics. A first significant example is given by solar neutrino studies, a sector in which reactor experiments already gave an essential contribution, with KamLAND. As we are going to discuss in Section 4.4, it should be possibile at JUNO to study the medium and high energy part of the solar neutrino spectrum (namely the $^7$Be and $^8$B contributions), searching for important contributions to the solution of the solar metallicity problem and testing, at the same time, the consistency of the the standard LMA oscillation solution, especially in the vacuum to matter transition region. Eventual deviations from this oscillation pattern could be hints of the existence of Non Standard Neutrino Interactions.

The list of other particle and astroparticle topics that will be studied at JUNO includes the search for neutrinos originated by the explosion of an eventual nearby Supernova and the measurement of the diffuse Supernova neutrino background, the geoneutrinos and atmospheric neutrinos studies and the search for exotic phenomena (like the proton decay) or for signals of Lorentz Invariance Violation. All of them will be briefly discussed in the remaining part of this Section and we will show that also in these fields a reactor neutrino experiment, like JUNO, will make possible a significant step forward with respect to the present state of art.

### 4.1. The JUNO Detector Main Features

The JUNO detector [42,102] consists of twenty thousand tons of a liquid scintillator (LS), the linear alkylbenzene (LAB), chosen for its excellent transparency, high flash point, low chemical reactivity, and good light yield, with the addition of 2.5 g/L of 2,5-diphenyloxazole (PPO) and 3 mg/L of p-bis-(o-methylstyryl)-benzene (bis-MSB) as wavelength shifter. The scintillator is contained in a spherical acrylic inner vessel (made up by 260 panels with 12 cm thickness) of radius of 17.7 m and will be the largest liquid scintillator detector ever built. The whole structure is sustained by a stainless steel shell. The vessel will be surrounded by a double system of photomultipliers (PMT): 18,000 large (20-inch) and 25,600 small (3-inch) PMTs, facing the inner vessel, located in a water buffer, which, together with the central LS sphere, will form the so-called central detector. The latter will be immersed in an instrumented water pool serving as a muon veto detector.

The key features, essential for the success of the experiment are the following.

- The medium baseline, settled at the value (53 km) ideal for the mass ordering analysis, as discussed in Section 3.
- The very good energy resolution, reaching the level $\frac{\sigma(E)}{\sqrt{E}} \simeq 3\%$, essential to discriminate the spectrum wiggles and perform an efficient statistical analysis. In order to satisfy this requirement, it's important to take under control also the non-stochastic contribution to the energy resolution, that must be reduced below 1%. This unprecedented level of accuracy can be reached thanks to the optimal detector coverage by the PMT (about 75 %), their vey high light yield ($10^4$ photon/MeV) and the good value of the attenuation length (>20 m for a wavelength of 430 nm).
- The use of small PMTs, that can operate in photon-counting mode, in addition to the large ones, guarantees an improved systematic control and an increase of the dynamic range, useful mainly to treat potential large signals, like in case of Supernova neutrino detection.
- The cosmogenic background reduction. The overburden (about 700 m) guarantees by itself a significant reduction of the cosmic ray fluxes. Moreover, the pool containing 35 ktons of ultrapure water, instrumented with 2400 PMTs, in which the central detector is immersed, offers a shield against the natural radioactivity from the rock and the neutrons from cosmic rays and an efficient veto to cosmic-ray muons. The muon veto system includes also a top tracker composed by three layers of plastic scintillators.

- The use of a near detector, named TAO (Taishan Antineutrino Observatory)[3], is very important for a better detailed knowledge of the reactor antineutrino beam spectrum and of its time stability. As a matter of fact the "standard" reactor shape uncertainties have a minor impact on the mass ordering sensitivity, but, in any case, a continuos monitoring of the flux is important because in principle the reactor spectrum might have not yet observed micro-structures that could degrade the mass ordering sensitivity, by mimicking periodic oscillation patterns. The JUNO-TAO detector is a Gadolinium doped liquid scintillator detector, with a 1 ton fiducial volume, settled at a distance of 30 m from the reactor core. It will have a full coverage of SiPM (Silicon photomultipliers) that will operate at $-50\,°C$, in such a way to drastically reduce the dark noise.

### 4.2. Mass Ordering Study with JUNO Experiment

The main focus of JUNO and one of the most important contributions expected for the near future from reactor neutrino experiments to elementary particle and astroparticle physics is for sure the detailed study of neutrino mass ordering. As explained in Section 3, JUNO should be able to recover important information about this topic, by studying the hierarchy dependent corrections to the inverse $\beta$ decay spectrum and performing a global fit of these and previous neutrino experiment data, by means of a $\chi^2$ analysis performed for both the possible neutrino mass ordering. The difference $\Delta\chi^2 = \left|\chi^2_{MIN}(NH) - \chi^2_{MIN}(IH)\right|$ between the values of the $\chi^2$ minima obtained for the two possible mass ordering gives an indication of the discrimination power of the experiment.

Taking advantage from the large statistics, guaranteed by the huge detector mass, and from the very good energy resolution, it should be possibile to determine the mass hierarchy with a confidence level around $3 - 4\,\sigma$. As widely discussed in [42], in the realistic experimental configuration, for a resolution $\sigma(E)/\sqrt{E} \simeq 3\%$, it should be possible to reach after six years of data taking a value of $\Delta\chi^2 \simeq 9$. This value could be further increased, reaching $\Delta\chi^2 \simeq 16$, by using the information on the squared mass differences coming from present and future LBL accelerator data (mainly T2K and NO$\nu$A). For the exact statistical interpretation of this result we address the interested reader to the wide debate available in literature on this topic [104–108].

The JUNO data will offer an essential advantage with respect to the ones collected by the other experiments studying the neutrino mass ordering: by looking at the vacuum oscillation, JUNO does not suffer from the uncertainty on the Earth density profile, which is, instead, important for the neutrino telescopes studies, based on the matter effects in the atmospheric neutrino propagation. In addition to this, JUNO results are not influenced by the CP-violating phase ambiguity, which is the main source of uncertainty for the LBL analyses (reported in Section 3.2) and by the $\theta_{13}$ value and they are only mildly affected by the choice of the 3 or 3+1 flavors pattern. In the future, it should be possible also to combine the JUNO results with the ones coming by the neutrino telescopes, taking advantage not only by the obviously increased statistics, but also by the fact that a fake solution for one of the two kind of experiments will be strongly suppressed by the data of the other experiments.

### 4.3. Mass and Mixing Parameters Measurement

Another important issue at JUNO will be the precise measurement, at the subpercent level, of some mass and mixing parameters, namely $\theta_{12}$, $\Delta m^2_{21}$ and a combination of $\Delta m^2_{31}$ and $\Delta m^2_{32}$. In most cases the JUNO's result should guarantee an improvement of almost an order of magnitude, with respect to the present experimental accuracy, as summarized in Table 3. An improvement in the oscillation parameters determination is important, not only for a better knowledge of the mass and mixing pattern, and consequently for a discrimination between different possible theories beyond the Standard Model, but also for a correct evaluation of the potentialities of future experiments looking for effects whose amplitude is proportional to these parameters.

---

3　　For a more detailed description of JUNO-TAO project and detector see, for instance: [103].

**Table 3.** Expected accuracy with the use of JUNO data, compared with the present one, for the mass and mixing parameters for which a significant improvement is expected. The leading experiments for the present parameters determination are also reported. The JUNO expected accuracies are recovered by [42] and by the most recent collaboration's analyses.

| Oscillation Parameter | Current Accuracy [109] (Global $1\,\sigma$) | Dominant Experiment(s) | JUNO Potentiality |
|:---:|:---:|:---:|:---:|
| $\Delta m_{21}^2$ | 2.3 % | KamLAND | $\simeq 0.6\%$ |
| $\Delta m_{ee}^2 = \left[\cos^2(\theta_{12})\Delta m_{31}^2 + \sin^2(\theta_{12})\Delta m_{32}^2\right]$ | 1.8 % | MINOS, MINOS+, T2K | $\simeq 0.4\%$ |
| $\sin^2\theta_{12}$ | 5.8% | SNO | $\simeq 0.7\%$ |

*4.4. Solar Neutrino Physics at JUNO*

Even if it has been specifically designed to study reactor antineutrinos, the JUNO detector, with its very large mass and unprecedented high-energy resolution, can also contribute to a better knowledge of solar neutrinos, shedding light on some of the topics still to be clarified in this field.

Solar neutrinos have been widely studied in the past, by different kind of experiments, ranging from the radiochemical ones (Homestake [110,111], Gallex [112,113], SAGE [114,115] and GNO [116,117]) to the Cerenkov detectors (Kamiokande [118], SuperKamiokande [119–122] and SNO [21]) and, at last, to scintillators, with Borexino [123–127]. The results collected over almost fifty years of experimental studies and phenomenological analyses [128–130] made possible a fundamental progress in our knowledge, not only of the mechanism ruling the fusion processes inside our star, but also of some key points fundamental for all elementary particle physics.

In fact the solution of the long standing "Solar Neutrino Puzzle", based on the flavor oscillation mechanism and the interaction with matter [26–29], offered a smoking gun essential to prove that neutrinos are massive particles and to show the need to go beyond the usual version of Standard Model of electroweak interactions. A fundamental cross check of this solution came by the reactor antineutrino experiment KamLAND [19], which tested the same region of oscillation parameters, as described in Section 1. Moreover, the accurate determination, through the combination of the data from the various solar neutrino experiments, of the fluxes for the full solar neutrino spectrum and the opportunity of checking the mechanisms ruling the pp chain (that is the main fusion process taking place inside the Sun), made possible by simultaneous detections at Borexino [126,127] of all the neutrinos emitted in this process, gave a unique opportunity to test the validity of Solar Standard Models (SSM) [131] and contributed significantly to their improvement.

Nevertheless, some points still need to be clarified and there are some questions already waiting for an answer, that would be extremely relevant, both for astrophysicists and for elementary particle physicists. First of all, it would be very important to discriminate between the two possible versions of the SSM [132–134] and try to solve the so called Solar metallicity problem [128,129,135,136]. The main step forward towards the final solution of this problem, will probably come by the measurement, at Borexino or at some future experiment, of the CNO neutrino flux, for which the predictions by low-Z and high-Z Solar Standard Models are significantly different. Nevertheless, every improvement in the accuracy determination of $^7$Be and $^8$B neutrinos could contribute to the solution of this problem, also because it could be important to solve the ambiguity between high-Z and low-Z with modified opacity models[4]. Another relevant issue would be the measurement of the $^8$B neutrino spectrum in the region of vacuum to matter transition, with the aim to test the stability of the LMA oscillation

---

[4] About this topic see the reference [42] and the references therein and see also: [137].

solution, validating or definitely excluding the hypothesis of Non Standard Neutrino Interactions (NSI)[5], as already explained.

Once more a fundamental contribution to this research project could come by the reactor neutrino experiments and more specifically by JUNO. In particular, it should be possible to study at JUNO the contribution of the electron neutrino spectrum corresponding to $^8$B and $^7$Be, and probably also hep, neutrinos. The success of these analyses will require, in addition to the good energy resolution and the big statistics (that are the strengths of the experiment), also the capability of reaching levels of radiopurity at least partially comparable with the Borexino ones. Considering the full energy spectrum, up to 15–16 MeV, at JUNO it should be possible to achieve, in case of "ideal" radiopurity, a value around 2 to 1 for the signal-to-background ratio and, even if this ratio is partially reduced in the relevant energy range from 2 to 3–4 MeV, it should be possible in any case to test new physics models, taking advantage by the high statistics and the very good energy resolution of the experiment. For a more detailed discussion about JUNO sensitivity to solar neutrinos and mainly to the $^8$B and $^7$Be contributions we refer the interested reader, in addition to [42,146] and to a detailed study by JUNO Collaboration about this topic, which is in progress.

### 4.5. Geoneutrinos and SuperNova Neutrinos Measurements with a Reactor Experiment

A reactor neutrino experiment is from a certain point of view also the ideal experimental apparatus to study the so called geoneutrinos, that are the antineutrinos emitted by natural radioactive decays taking place inside the Earth. In fact, the experimental channel to look for signals of geoneutrinos is the inverse $\beta$ decay, that is the main process for which a reactor antineutrino is designed. At the same time, this is also the main problem to solve to extract the geoneutrino signal from the reactor signal, which represents its main background.

From this point of view, an experiment like JUNO will take advantage from its main characteristics: the huge mass, that guarantees an high statistics, the good radiopurity levels, the medium baseline (53 km), which determines a reduction of the reactor neutrino flux with respect to the SBL experiments, and, above all, the excellent energy resolution. This last point would help in discriminating the geoneutrino signal from the reactor background, because the geoneutrinos produced in the radioactive decay channels of $^{238}$U and $^{232}$Th[6] have an energy spectrum centered around values slightly lower than the ones of the reactor antineutrinos. The sensitivity to the geoneutrino signal could be particularly interesting during the first period of run of the experiment, when the power of some of the reactor cores could be lower than the designed one.

The geoneutrino signal measurement is very important in order to estimate the radiogenic contribution to Earth heat power and test Earth's different geochemical models. One year of JUNO data taking should be enough [147–149] to exceed the present number of geoneutrino events collected by previous experiments which performed similar measurements, that are KamLAND [150] and Borexino [151,152]. Analyzing together the geoneutrino data from all the three experiments, it should be possible to improve significantly the estimate of the Th and U abundance in the Earth, shedding light on the relative relevance of the radiogenic contribution to the heat flow of the Earth and contributing also to solve the long standing puzzle about the origin of Earth's heat [153].

Another relevant study that could be performed by JUNO [42,154,155], and more generically by an ultrapure liquid scintillator [156], is the detection of an eventual Supernova (SN) neutrino burst and of the diffuse SN background. Both issues could give answers to important physical and astrophysical questions, like the knowledge of the mechanisms ruling stars formation and evolution, the SN collapse and explosion and the related production of the heavy chemical elements.

---

[5] About the models predicting possible Non Standard Neutrino Interactions and the limitations to these models coming from phenomenology, see, for instance: [138–145].

[6] The antineutrinos emitted by $^{40}$K radioactive decays are not directly measurable, because their energy is below the threshold for inverse $\beta$ decay.

*4.6. Atmospheric Neutrino Studies at JUNO*

The study of atmospheric neutrinos with a detector like JUNO is a challenging, but interesting task. The main results in this field have been obtained by SuperKamiokande experiment [157,158], which, being a water Cerenkov detector, took advantage by the possibility of discriminating quite easily the $\nu_\mu$ from the $\nu_e$ induced signal. In order to perform a similar analysis in a scintillator experiment like JUNO, one has to develop some clever ad hoc experimental procedure.

Together with the flavor identification, one would like also to obtain a good energy reconstruction and background knowledge and rejection, especially for the most dangerous background, that is represented by cosmic muons, simulating $\nu_\mu$ induced events. In order to achieve such ambitious results, the scintillation light alone is not enough, but one can exploit different combined experimental techniques. First of all, it is possible to use the fact that the first PMT hit is associated with Cerenkov emission and can be used to reconstruct the lepton direction. As a matter of fact this idea works only for high energy through going events. A more promising opportunity is offered by a flavor identification based on a detailed study of the event time profile. In fact the $\nu_\mu$ and $\nu_e$ generated events are characterized by different light distributions and, in general, larger time profiles are expected for muon events. In addition to this, the idea is to take advantage from the very good JUNO's energy resolution and to use the collected scintillation light to recover a calorimetric information, making possible the precise measurement of the event's energy. The $\nu_\mu$ generated events (above $\simeq 7$ GeV) should pass through the detector and, therefore, in these cases one will have to find a clever energy reconstruction algorithm, working for up-going through passing events. For contained high energy $\nu_e$ generated events there could be, on the opposite, a problem of large PMT saturation (presumably starting from about 10–20 GeV). The idea in this case would be to complement the information with the one collected by small PMTs, that have a worst energy resolution, but should not present saturation problems.

An efficient reconstruction algorithm based on a probabilistic unfolding method has been successfully developed [159], in order to infer the primary neutrino energy spectrum by looking at the detector output. The simulated spectrum has been reconstructed between 100 MeV and 10 GeV, showing a great potential of the detector in the atmospheric low energy region.

*4.7. Search for LIV Signals and Other Exotic Studies at JUNO*

A multiple-purpose experiment like JUNO offers also the possibility of searching for "exotic" still unseen processes, forbidden in the Standard Model and which would be indications of new physics, like the proton decay [42] (that will be investigated via the $p \to K^+ \bar{\nu}$ decay channel), and testing fundamental properties, like the Lorentz symmetry invariance.

The relevance of Lorentz invariance, one of the grounding symmetries at the basis of relativistic theories, justifies the great attention dedicated by many theoretical and experimental studies to more and more accurate tests of its validity. In the recent past potential sources of Lorentz Invariance Violation (LIV) have been advocated also as possible explanations of partial deviations of the experimental data (mainly about high energy cosmic rays) from the expected pattern[7]. Moreover, there are consistent indications that LIV could be naturally incorporated in quantum gravity theories [161,162]. The kind of Lorentz invariance violation (LIV) signals usually considered in literature are proposed having in mind the theoretical framework of the so called "Standard Model Extension" [163,164] and they are associated to non isotropic effects, like the signals of spectral distortion and of sidereal variations [165]. These kind of studies can be performed also at JUNO [42,165], but, in addition to them, there is a new interesting opportunity that could be investigated.

---

[7] For a review on this topic, see for instance: [160].

In a recently proposed model [166,167], denoted as HMSR (Homogeneously Modified Special Relativity), some sources of LIV are introduced, starting from the modification of the energy-momentum dispersion relation, with a geometrical origin (by using the Finsler geometry) and in such a way to preserve the space time isotropy and build an isotropic CPT conserving extension of the Standard Model. The potential experimental effects induced by this kind of LIV would be represented by isotropic corrections to the usual known phenomenology. In particular the dispersion relation modification would induce a change in neutrino propagation and, consequently, some corrections to the usual oscillation behaviour. In addition to the usual leading term, proportional to $\frac{L}{E}$, a subleading correction term, proportional to $L \times E$ would appear [166].

The way to test the existence of this kind of corrections is to look for signals of isotropic corrections to the oscillation pattern emerging for long baselines and high energies. The ideal experimental framework is clearly represented by the study at neutrino telescopes of high and very high energy cosmic neutrinos; one can also think of the study of ultra high energy cosmic neutrinos in case of emission by very far sources. The extension of this kind of studies to the observation of high energy atmospheric neutrinos at JUNO represents another potential field of analysis that is presently under investigation [168,169]. The first preliminary results are interesting, but the feasibility of such a research project is strictly connected with the development of the techniques, described in Section 4.6, to study the atmospheric neutrino signal with a liquid scintillator like JUNO and to their extension to the multi-GeV energy region.

The possibility of testing Lorentz invariance and searching for LIV signals by means of neutrino experiments is an important example of studies aiming to analyze the impact on neutrino oscillations of gravitational effects. This topic has been widely debated in literature [170–173][8]. It was already considered, for instance, in the study of solar and atmospheric neutrino data, for what concerns the possibility of gravitational induced effects that could modify the "traditional oscillation pattern", with the introduction of additional quantum mechanical phases [175,176]. The analysis has been extended to the study of these potential effects for neutrinos emitted by far astrophysical objects, with particular attention to the case of strong gravitational fields [177,178]. It is worthwhile to also recall the recent studies about neutrino oscillations in extended gravity theories [179] and the ones about the dependence upon the neutrino mass hierarchy of the potential effects of neutrino lensing induced by gravitational sources [180].

## 5. Discussion and Conclusions

In this paper, we reported the main aspects of reactor antineutrino experiments, recalling and discussing the important results that have been obtained in this research field and the great impact they had on the development of elementary particle physics, astrophysics and geological studies.

Then, we focused our attention on a change of paradigm that is taking place. In the near future, the short baseline reactor experiments will continue to play an important role, with studies aiming mainly to find signals in favor of the sterile neutrino existence or to definitely disprove this theoretical hypothesis. However, the main novelty will be represented by a new kind of analyses, that will be developed by means of medium baseline reactor experiments and whose main goal is the determination of the neutrino mass ordering, an important open issue of great interest both from the theoretical and the experimental point of view.

The possibility of performing this kind of studies was preconized already in the eighties, in [97], but the feasibility of this research poject was confirmed only in 2012, when the three SBL experiments (Daya Bay, RENO and Double CHOOZ) proved definitely that the $\theta_{13}$ mixing angle is significantly different from zero and, therefore, by studying the inverse $\beta$ decay of reactor antineutrinos in medium

---

[8] For a non exhaustive, but rich and interesting discussion about the impact of quantum gravity effects on neutrino oscillations, see also: [174].

baseline experiments characterized by huge detectors and extremely high energy resolution, it is possible to investigate the mass ordering-dependent effects, that are proportional to $\sin^2 \theta_{13}$ (like in the case of CP violating effects). The first kind of reactor antineutrino experiment aiming to reach this goal, the liquid scintillator experiment JUNO, is almost ready for data collection and will become operative in the very next years in China.

We discussed in detail the main JUNO characteristics and potentialities, considering different aspects of its rich research program, which is not limited to the study of reactor antineutrinos, but covers also the mass and mixing parameters determination, the study of solar neutrinos, geoneutrinos and many other topics, confirming once more the very fruitful interplay between reactor experiments and all the other sectors of neutrino phenomenology of physical and astrophysical interest.

Additional information about the possible future synergies between reactor antineutrino experiments and LBL accelerator experiments, neutrino telescopes and other future neutrino experiments can be found also in [75].

**Author Contributions:** Conceptualization: V.A. , L.M., G.R. ; methodology V.A., L.M., G.R. ; software, data  curation and analytical calculations V.A.; validation: V.A., L.M. , G.R.; formal analysis V.A., L.M.; investigation V.A. ; writing—original draft preparation V.A. ; writing—review and editing V.A., L.M. and G.R. All authors have read and agreed to the published version of the manuscript.

**Funding:** This research received no external funding.

**Acknowledgments:** The authors would like to thank all the colleagues and friends of JUNO Collaboration, and  mainly the members of the JUNO publication committee for very useful suggestions and comments.

**Conflicts of Interest:** The authors declare no conflict of interest.

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
