# Peer review of "Present and Future Contributions of Reactor Experiments to Mass Ordering and Neutrino Oscillation Studies"

_universe, doi:10.3390/universe6040052_

Round 1

Reviewer 1 Report

I have read the review paper of Antonelli, Miramonti, Ranucci.

I collected a few recommendations that should be readable in the annotated PDF and I summarize here below.

My only additional suggestion of some relevance to authors is to emphasize which are the new points of this review work in particular in comparison with their ref. [24], that in my view should be cited before.

abstract

The text mentions "The first example of this kind," but KamLAND is in the same class.

page 2

1) I advise to add mention to Fermi, as w/o his quantitative theory, the hypothesis of Pauli is not enough to predict the existence of inverse beta decay.

2) a more detailed discussion of the uncertainties of reactor antineutrino flux is necessary.

page 5

I would distinguish a bit better the new experiments from their theoretical interpretation, and moreover I would add references on that.

page 6

(two misprints)

page 7

1) I suggest to cite the Delta chi^2 obtained.

2) I recommend to check whether atmospheric data are included in global analyses.

page 8

(one misprint)

one recommendation as per the papers cited in ref.[55], possibly putting them in historical order (rather that in reverse order). If more discussion is needed, please add it.

page 10

1) fig.4 needs editing.

2) a claim concerning NSI  sounds to me unclear and seems to be premature.

page 13

1) the issue of metallicity need to be distinguished better from the one of metallicity.

2) the possible specific contribution of JUNO to metallicity, if any, needs to be discussed. The priority of this goal needs to be discussed.

3) I advise to cite the original papers on NSI.

page 14

1) more discussion on background is advisable.

2) I suggest to add a figure to compare the possible contribution to geoneutrino physics, and to compare it with KamLAND and Borexino results.

3) (I suggest a paper appeared before your ref.81).

4) it would be useful to mention which reactions can be studied with atm. neutrinos.

5) in the study of atm. neutrinos, I think that the low energy and high energy regions should be distinguished (also in view of the physics reach). Note that SuperKamiokande has the problem of muons below the Cherenkov threshold, while JUNO does not.

page 15

1) I recommend to resume the motivations for the the LIV.

2) a more precise comment concerning the LIV effect would be desirable.

page 16

I am confused by the last statement of the text.

Author Response

Dear referee,

We read carefully your report and we really thank you for your message and your detailed and careful suggestions of modifications and additional analyses. 

We prepared a revised version of our paper keeping into account your remarks and the analogous ones by the other referees. 

We are going to resubmit this revised version. 

Here attached you can find a pdf file containing our point-by-point answer to your remarks. 

Best regards, 

The authors. 

Reviewer 2 Report

This manuscript by Antonelii et al. is a good overview of neutrino
oscillation physics. But it has some serious problems that require
corrections.

The manuscript has some misspellings on lines 530 and 531.
Line 126 suggests a depth of 900 hundred mwe, which would be 90 km (90000 m)

The manuscript is still unfinished as mentioned on line 192 "Subsection still
to be developed"

Line 156 suggests a problem with the predictions of the u 235 to the neutrino
spectrum. Please check this. I thought the U 235 was well understood and it
was the U 238 that was poorly known. But things may have changed.

Figure 3 is a plot of "arbitrary units" as a function of L/E.
But the Nonoscilated curve does not depend on L/E. Perhaps the curve can be
redrawn or better explained. The vertical scale has some significance.
Is this actually a plot of the 1/E distribution at the JUNO location?
Is L fixed?

Many statements are made without detailed exposition or proof.
Some additional details would strengthen the paper.

Author Response

Dear referee,

We read carefully your report and we thank you for your message and your suggestions of modifications and additional analyses. 

We prepared a revised version of our paper keeping into account your remarks and the analogous ones by the other referees. 

We are going to resubmit this revised version. 

Here attached you can find a pdf file containing our point-by-point answer to your remarks. 

Best regards, 

The authors. 

Reviewer 3 Report

This paper is a review paper not for a specific new study. It is well written with good clarification but four general comments:

1) There should be more discussion and comments on the sterile neutrino studies in reactor neutrino experiments for both short baseline and very short baseline reactor experiments. The sterile neutrino results from accelerator neutrino experiments should be mentioned as well.

2) Please illustrate the impact of oscillation parameter pre-fit uncertainties on the mass ordering measurement in JUNO.

3) Quite some pages talk about JUNO potentialities in studying geo-neutrinos, solar neutrinos (for solar model), supernova neutrinos, and Lorentz invariance violation, etc.

I think these are beyond the scope of this paper entitled "Present and future contributions of reactor experiments to mass ordering and neutrino oscillation studies" . These studies are not much related to "mass ordering and neutrino oscillation". If we expand the scope (title) of this paper to such studies in reactor experiments, the results from other experiments not just JUNO should also be reviewed. 

4) All figures should be explicitly cited and compiled with a paper-level resolution. 

Several detailed comments:

1) Line 167: in States -> in the United States.

2) What is line 192 (the first sentence for section 3.1)? Should be removed.

3) Line 287: please use more complete and accurate language instead of "1-2 sector"

4) Line 315: Linear -> linear

5) Table 3 results are not "current". Please update accordingly based on the most recent results.

Author Response

Dear referee,

we read carefully your report and we thank you for your message and your suggestions of modifications and additional analyses.

We prepared a revised version of our paper keeping into account your remarks and the analogous ones by the other referees.

We are going to resubmit this revised version.

Here attached you can find a pdf file containing our point-by-point answer to your remarks.

Best regards,

The authors.

Reviewer 4 Report

The work is useful for readers and well written. The topic is hot and I think it deserves publication, albeit I've just a couple of comments that the authors might fulfill in the improved version.

1) Space physics is currently a hot topic of theoretical physics. In particular, space laboratories will be useful, in ideal experiments, to propose to check new physics. So, to heal the problem of mass hierarchy, the authors think  that ideal experiments in space could be realized? And how?

In other words, in which way space experiments must be built up? I mean: the authors can suggest how an ideal experiment in space can be built up to address the issue of hierarchy to give reasonability to the work.

2) The fact that neutrino oscillation is influenced by strong field gravity is not cited enough. I suggest to put somewhere the fact that oscillation is also influenced by gravity. As seminal papers I simply suggest: A. Geralico, O. Luongo, Physics Letter A, 376 (2012) 1239-1243; D.V. Ahluwalia, C. Burgard, Gen. Rel. Grav., 28 (1996) 1161 and D.V. Ahluwalia, C. Burgard, Phys. Rev. D 57 (1998) 4724.

After these changes, I would like to give a final look at the manuscript to be sure that everything is fine.

Author Response

Dear referee,

we read carefully your report and we thank you for your message and your suggestions of modifications and additional analyses

We prepared a revised version of our paper keeping into account your remarks and the analogous ones by the other referees

We are going to resubmit this revised version

For what concerns in particular your remarks, we added at the end of Section 4 (Subsection 4.7) quite a wide discussion, also with a certain number of references, about gravitational effects and neutrino oscillation. 

Best regards

The authors

Round 2

Reviewer 1 Report

I thank the Authors for taking into consideration the advice and suggestions.

I am pleased to congratulate them for the care taken in improving the text of their work, and to be able to recommend its publication as it stands.

Reviewer 3 Report

Thanks for your revisions.

The present version is of good clarity and completeness. I accept it a publication in the present form.

Reviewer 4 Report

The paper is excellent. I do recommend it for publication.